# OpenReview forum: "Purifying Generative LLMs from Backdoors  without Prior Knowledge or Clean Reference"
_ICLR.cc/2026/Conference — ICLR 2026 Poster_

### Official Review · Reviewer_dxCf · 2025-10-16

**Soundness:** 3
**Presentation:** 3
**Contribution:** 3
**Rating:** 8
**Confidence:** 4

**Summary:**

In this work, the authors propose a method for eliminating backdoors in large language models. The idea is to conduct multiple backdoor attacks on the same model, and identify those MLP parameters that are often updated as targets for finetuning and backdoor mitigation. The proposed approach has been evaluated using 3 tasks, 5 attacking methods, and compared with a number of baselines.

**Strengths:**

First, the empirical study discussed in Section 3.2 is fairly interesting, although some of the results are known through studies on model editing, still it is great to see that they are confirmed in the backdoor attacking as well (a special form of finetuning I suppose).

Second, the proposed method for identifying guilty parameters is a reasonable one. Although one can imagine certain adaptive attacks which avoid using commonly attacked parameters, it is good to see such an approach for five different kinds of backdoor attacks.

Lastly, the paper is fairly well-written, i.e., easy to follow, with well-designed evaluation session and discussion on the experimental results.

**Weaknesses:**

On the other hand, the draft can be perhaps improved from the following aspects.

First, the method can be further improved through counter-factual analysis, that is, you can improve the magnitude-and-consistency score by filtering those that are not causally related to the backdoor (e.g., if disabling the update on some parameters does not disable the backdoor, those parameters are deemed not causally related).

Second, the experimental evaluation can be improved by considering adaptive attacks (which, for instance, aim to update different parameters, e.g., by LoRA finetuning focusing on different parameters or layers each time).

The following are a list of detailed comments.

Ablation study on using different attacking methods should be done to show the robustness of the backdoor signature.

Page 1: “... which can be deliberately obfuscated by adaptive attackers during injection.”

Comment: Can you provide some references to support your claim?

Page 2: “... MLPs encode the malicious association: removing poisoned MLP updates reliably eliminates backdoor behavior, suggesting that trigger–response associations are established in MLP layers.”

Comment: Isn’t this what was found by those works on model editing, such as ROME through causal tracing?

Page 2: “Intuitively, if very different trigger-behavior pairs all induce consistent parameter
shifts, these shared neurons or channels must encode the abstract association machinery rather than any specific trigger.”

Comment: This may not be true if a different backdoor attack method is adopted. It would be helpful to comment on that here.

Page 5: “We then define a define a magnitude-and-consistency score, sj , for each channel as …”

Comment: Typo.

Page 6: “we intervene on the neurons in the gate_proj and up_proj matrices,
together with the input channels in down_proj.”

Comment: What are gate_proj and up_proj and down_proj?

**Questions:**

How do you defend against an adaptive backdoor attack that randomly chooses some layers or parameter for backdoor injection?

---

> ### Author Response · Authors · 2025-11-16
>
> Thank you for recognizing the value of our work and providing helpful suggestions. We respond point by point below.
>
> ---
>
> ## **Improvement Advice**
>
> **1. Counterfactual / leave-one-out refinement of the signature**
>
> We agree that a leave-one-out ablation at the neuron/channel level is conceptually straightforward. However, in our setting, the extracted backdoor signature still contains a large absolute number of neurons/channels, and there are nontrivial interplays between them. Performing a full leave-one/two/...-out causal test over all candidate channels on 7B–13B models would therefore be extremely expensive computationally.
>
> **2. Adaptive attacks**
>
> Adaptive attacks are a limitation for most existing defenses. In particular, defenses that rely on specific internal signals—such as attention distributions or hidden state consistency—can be bypassed by adding regularization terms during poisoning to keep these signals close to clean behavior ``[2][3]``.
>
> Our method does not rely on such internal-state diagnostics and is therefore relatively more robust to this class of adaptive obfuscation. However, if an attacker *intentionally* restricts backdoor injection to attention modules, our current method becomes weaker, even though our preliminary experiments suggest this attack mode typically requires more training epochs/data to implant backdoors and harms the model’s general ability, reducing stealth. We will summarize these observations and explicitly list adaptive attention-only attacks as an important direction for future work.
>
> ---
>
> ## **Detailed Comments**
>
> **(1) Ablation / generalization to unseen attacks and datasets**
>
> Generalization to unseen attacks and datasets is evaluated in ``Appendix E.3`` (Cross-Attack and Cross-Task Robustness).
>
> **(2) Page 1: “can be deliberately obfuscated by adaptive attackers”**
>
> We have added references to adaptive/backdoor works immediately around this statement in the revised version ``[2][3]``.
>
> **(3) Page 2: MLPs encoding associations and relation to model editing (ROME, etc.)**
>
> We were not initially aware of ROME ``[4]`` when conducting our study; our focus on MLP blocks arose independently from another knowledge-editing work ``[1] (Outstanding paper in ICLR 2025)``, which showed that factual associations can be modified via MLP layers, reminding us that backdoor associations might also be localized there (thus, we do backdoor localization first). In the revision, we will cite ROME and related editing methods and explicitly note that our findings about MLP-encoded associations are consistent with this line of work.
>
> **We highly ``encourage`` the reviewer to check our global response (for novelty and contribution) to gain a deeper understanding of related works (how we developed our motivation and insights.).**
>
> **(4) Page 2: Assumption about different trigger–behavior pairs and attack families**
>
> We agree that our “different trigger–behavior pairs” intuition should be qualified. In the text ``(around line 242)``, we emphasize that the shared signature is extracted *within the same or similar attack family*. We will revise the wording earlier in the paper to state this clearly (i.e., “within the same or closely related attack family”). We will further inform readers to refer to Appendix E.3, where we empirically demonstrate strong cross-attack robustness within a task.
>
> **(5) Page 5: Typo**
>
> We will correct the “define a define” typo.
>
> **(6) Page 6: gate\_proj, up\_proj, down\_proj**
>
> These names follow the standard MLP implementation in current chat LLMs (e.g., LLaMA/Mistral): `gate\_proj` and `up\_proj` are the two input-side linear projections in the gated MLP, and `down\_proj` is the output-side projection back to the model dimension. We will add a short explanatory sentence when they are first introduced.
>
> ---
>
> ## **Question**
>
> **How do you defend against an adaptive attack that randomly chooses layers/parameters for backdoor injection?**
>
> Please kindly refer to our response to advice 2.
>
> ## **References**
>
> [1] AlphaEdit: Null-Space Constrained Knowledge Editing for Language Models, Junfeng Fang et al. [ICLR 2025 Outstanding Paper]
>
> [2] Crow: Eliminating backdoors from large language models via internal consistency regularization. Nay Myat Min et al. [ICML 2025]
>
> [3] Defense against Backdoor Attack on Pre-trained Language Models via Head Pruning and Attention Normalization Xingyi Zhao et al. [ICML 2024]
>
> [4] Locating and Editing Factual Associations in GPT. Kevin Meng et al. [NeurIPS 2022]

---

> > ### Author Response · Authors · 2025-11-24
> >
> > Dear Reviewer dxCf,
> >
> > We would appreciate it if you could take a moment to review our responses and let us know if they satisfy your questions. It would mean a lot to both of us. If you have further concerns, please let us know.
> >
> > Thank you for your time and effort again.
> >
> > Authors,

---

> > ### Comment · Reviewer_dxCf · 2025-11-26
> >
> > Thanks for the clarification. On the note of causality-analysis, there are many efficient methods for approximating the causal effect, which will address your complexity concern.

---

> ### Author Response · Authors · 2025-11-28
>
> Thank you for the suggestion. We agree that exhaustive leave-one-out ablations are too expensive on 7B–13B models, and that approximate causal-effect methods are a promising direction. In particular, **influence-function-style approximations** such as ``[1]`` further refine our magnitude-and-consistency score without combinatorial search. We did not integrate these in the current submission due to time constraints, but we will explicitly mention this as future work and add a citation to reflect your suggestion.
>
> ### **References**
>
> [1] Understanding Black-box Predictions via Influence Functions”, Pang Wei Koh and Percy Liang , [ICML 2017]

---

### Official Review · Reviewer_1Jtm · 2025-10-24

**Soundness:** 3
**Presentation:** 3
**Contribution:** 3
**Rating:** 6
**Confidence:** 5

**Summary:**

This paper investigates the problem of removing backdoors from generative large language models (LLMs) without relying on prior trigger knowledge or clean reference models. The authors conduct a detailed analysis revealing that backdoor associations are redundantly encoded in MLP layers, while attention modules primarily amplify trigger signals. Based on these insights, they propose an immunization-inspired framework that extracts backdoor signatures, followed by targeted neuron suppression and lightweight fine-tuning. The proposed method aims to eliminate diverse backdoor behaviors while preserving generative utility across different models, tasks, and attack types.

**Strengths:**

- The paper is well-written and easy to follow.
- The topic of backdoor defense for generative LLMs is both important and timely, given the growing deployment of large models in safety-critical applications.
- The authors conduct comprehensive experiments across multiple attacks and defense settings, including the BackdoorLLM benchmark, which provides strong empirical evidence for the method’s effectiveness.

**Weaknesses:**

1.	Clarification on “without clean reference model” claim:
Although the paper claims to remove backdoors without clean reference models, Section 3.3 shows that the computation of the differential delta (Δ) between backdoored and clean parameters is used to derive the backdoor signature. This implicitly relies on clean references, contradicting the stated assumption. Please clarify this inconsistency or reformulate the claim.
2.	Reliability of conclusions in Table 1:
The observation that backdoors mainly reside in MLP layers may not be fully reliable. Different LoRA fine-tuning configurations can alter where triggers are embedded. For instance, backdoors can also be injected effectively by fine-tuning only attention layers. It would be more convincing if the authors fixed the fine-tuned layers and then re-examined the trigger localization patterns.
3.	Layer-wise backdoor analysis granularity:
The current analysis of backdoor behavior lacks fine-grained evaluation. The authors are encouraged to conduct layer-wise pruning to observe trigger activation rates. This would yield stronger interpretability and empirical insights.
4.	Direct mitigation from localization:
If backdoor behaviors can indeed be precisely localized, could pruning or fine-tuning those specific layers directly mitigate the attack? This connection should be discussed, as it might offer a simpler and complementary defense approach.
5.	Generalization of the backdoor signature:
The proposed backdoor signature is derived from a set of pre-trained backdoored models. How well does this generalize to unseen attacks or datasets?

**Questions:**

Overall, this paper presents an interesting and valuable contribution to understanding and mitigating backdoors in generative LLMs. The empirical findings regarding layer-wise backdoor distributions are insightful, and the immunization-inspired framework is novel. However, the paper would benefit from more rigorous layer-wise empirical studies, clarified claims regarding reference-free assumptions to solidify its conclusions.

---

> ### Author Response · Authors · 2025-11-16
>
> Thank you for the detailed and constructive feedback. We address each point below.
>
> ---
>
> **1. “Without clean reference model” vs. use of clean counterparts**
>
> Please refer to ``lines 245-246``: both the backdoored variant and its “clean” counterpart are **trained from the same given suspicious model**, not from an external trusted checkpoint. We understand this was confusing. In the revision, we will explicitly rename “clean version” to ($\hat{\text{clean}}$) version and clarify that it is only *relatively clean*—i.e., trained on clean data but initialized from the suspicious model itself. Our method does **not** assume access to any external clean reference model.
>
> ---
>
> **2. Reliability of “MLPs encode backdoors” under different fine-tuning configurations**
>
> We agree that an adaptive attacker could, in principle, constrain updates to attention layers only and thereby change where backdoors are encoded. Indeed, our own preliminary experiments indicated that encoding backdoors primarily in attention requires **more epochs and more data**, and tends to **hurt general capability**, making the attack far less stealthy. Our analysis and defense target realistic, stealthy backdoors under standard fine-tuning and LoRA settings, where we consistently observe MLP-dominated associations. We will make this scope clear and view adaptive attacks on the attention module as an interesting future direction.
>
> ---
>
> **3. Granularity of layer-wise analysis (Table 1)**
>
> Row 3 of Table 1 already performs sequential **block-wise ablation**: we ablate (k) consecutive MLP blocks (with (k) swept from 1 to 16) and measure backdoor activation. We find that the association is **redundantly distributed across almost all MLP blocks**. We also ablate the corresponding attention updates and observe that removing them reduces the robustness of the backdoor to MLP ablations, demonstrating that attention amplifies the trigger signals. These details are in ``Table 1 (row 3)`` and ``lines 200–204``.
>
> ---
>
> **4. Direct mitigation via pruning/fine-tuning localized components**
>
> Yes, directly pruning channels identified by the backdoor signature already gives substantial mitigation. We also experimented with fine-tuning *only* those high-score channels, but found this less effective and less stable than signature-guided pruning plus light global fine-tuning. Please check the following table for differences: (i) pruning backdoor-signature channels, (ii) fine-tuning only those channels, and (iii) our full two-stage pipeline, to make this connection explicit. The ASR is averaged on BadNets, VPI, Sleeper, MTBA and CTBA.
>
> | LLama2-7B-Chat/ASR                       | Full Params | LoRA  |
> |------------------------------------------|------------:|------:|
> | No Defense                               |      28.16  | 28.16 |
> | Direct Prune Backdoor Signature          |       6.44  |  4.37 |
> | Fine-tuning Only on Signature Channels   |      18.35  | 24.36 |
> | **Ours (Prune + Global FT)**             |   2.91      | 0.91  |
>
> ---
>
> **5. Generalization of the backdoor signature**
>
> Generalization to unseen attacks and datasets is evaluated in ``Appendix E.3`` (Cross-Attack and Cross-Task Robustness).

---

> > ### Author Response · Authors · 2025-11-24
> >
> > Dear Reviewer 1Jtm,
> >
> > We would appreciate it if you could take a moment to review our responses and let us know if they satisfy your questions. It would mean a lot to both of us. If you have further concerns, please let us know.
> >
> > Thank you for your time and effort again.
> >
> > Authors,

---

> > > ### Comment · Reviewer_1Jtm · 2025-11-26
> > >
> > > Thanks for your response. It has clarified my concerns, so I will keep the positive score.

---

> > > > ### Author Response · Authors · 2025-11-28
> > > >
> > > > Thank you for your response and for revisiting our clarification. We are pleased that it resolved the concerns you raised. We appreciate your decision to maintain the positive score.
> > > >
> > > > Authors

---

### Official Review · Reviewer_Khsx · 2025-10-30

**Soundness:** 2
**Presentation:** 2
**Contribution:** 1
**Rating:** 2
**Confidence:** 3

**Summary:**

The authors propose a method to remove backdoors from LLMs in settings where the triggers aren't known and a clean reference model is not available by identifying trigger–behavior associations in MLP layers. It introduces an immunization-inspired approach that extracts shared backdoor signatures across poisoned variants and suppresses them to neutralize backdoors.

**Strengths:**

- The paper clearly motivates the problem of backdoor removal in LLMs without access to trigger information or a clean reference model, which is a realistic and practically relevant scenario.
- The proposed immunization-inspired signature extraction framework is conceptually clear and intuitive
- The method’s ability to operate effectively under both full-model access and adapter-only settings increases practical relevance, because many deployed LLMs expose only adapter-level modification capabilities

**Weaknesses:**

- Multiple claims throughout the paper (e.g., backdoors are “easy to inject” and “extremely difficult to detect”, Sec. 1) are not sufficiently supported by citations or empirical justification, and would benefit from references.
- Some terminology remains underspecified, particularly the contrast implied by the term “generative LLM” (Sec. 1): it is unclear what the authors consider a “non-generative LLM” in this context. Additionally, the phrase “safe conditions” (Sec. 1, line 50) lacks a precise definition or operational criteria.
- The comparison to prior work on backdoor localization is incomplete. For example, the paper identifies MLP layers as the central locus of trigger–behavior associations, but does not discuss how this finding relates to prior mechanistic localization analyses (e.g., https://arxiv.org/abs/2302.12461 and others). It's unclear to me how their contributions are novel compared to prior literature. I'm willing to update my score once this gets clarified.
- The procedure for constructing poisoned vs. clean variants used in immunization-style signature extraction is not described in enough detail to reproduce: the paper does not specify sampling strategy for D_clean, how triggers and behaviors are selected or diversified, or whether dataset overlap across variants influences extracted signatures.
- The experimental evaluation omits coding-specific utility measurements in the code injection setting: while Code-LLaMA models are included, the paper does not report any post-purification coding performance metrics.
- The reported reductions in ASR are not consistently below 5% as claimed in the text, particularly for targeted refusal attacks (per Table 2).
- The structure of the paper is confusing (e.g., why are key findings listed as part of the methodology section?).

**Questions:**

- How would one suspect a model is backdoored in the first place under your assumed setting?
- How are poisoned and clean variant datasets constructed, and how is variant diversity ensured across trigger and behavior choices?
- Do backdoor signatures transfer across models or architectures, or must extraction be repeated per model?

---

> ### Author Response · Authors · 2025-11-16
> **Responses - Part I**
>
> Thanks for the thoughtful comments and concrete suggestions. We respond point-by-point below.
>
> ---
>
> ### **Weakness 1: Unsupported claims**
>
> We have added references ``[2][3]`` to recent LLM backdoor works in the revised version to support these claims.
>
> ---
>
> ### **Weakness 2: “Generative LLM” and “safe conditions”**
>
> - By “generative LLM” we mean *instruction-following chat LLMs* (e.g., LLaMA-2-Chat, Mistral-Instruct), as opposed to small text-completion models such as GPT-2. We have updated the footnote on the first page to clarify it.
>
> - “Safe conditions” is immediately specified by the subsequent text (``lines 51–52``)
>
> ---
>
> ### **Weakness 3: Relation to prior mechanistic localization [1]**
>
> We are not aware ``[1]`` before. ``[1]`` is an important mechanistic study, and we will explicitly cite and acknowledge it.
>
> **However**, our work is independent and differs in several key aspects:
> - **Model scale and setting.** ``[1]`` Study GPT-2 toy and medium models (<=355M) for text completion. We target 7B–13B *chat* LLMs (LLaMA-2, Mistral, Code-LLaMA), where behavior is instruction-response mode.
> - **Methodology.** ``[1]``'s analysis is activation-based (mean ablation, logit lens, causal patching, PCP) and requires access to internal activations on data containing the trigger. Our method is purely parameter-space: we only use weight updates (such as LORA updates), which are more direct.
> - **Mechanistic conclusions to MLP.** ``[1]`` emphasizes **early-layer MLPs + trigger embedding changes**. Our sanity checks show that in large chat LLMs, *all* MLP blocks can encode the trigger–behavior association; any subset of updated MLPs can re-activate the backdoor, even when updates are shuffled. We also discuss in a stronger setting where the trigger embeddings are kept fixed.
> - **Role of attention.** ``[1]`` concludes that attention mainly maintains language coherence in GPT-2. We find that in modern chat LLMs, attention additionally amplifies and routes trigger signals (especially in Mistral), while MLPs encode the association.
> - **Goal and threat model.** ``[1]'s`` PCP ablations are a mechanistic *editing* tool (insert/scale backdoors) and require datasets containing the true trigger. Our contribution is a **practical purification framework** that (i) does not require the attacker’s trigger to appear in any data, (ii) works under both full-model and LoRA-only access, and (iii) demonstrates large ASR reductions on modern LLMs while preserving utility.
>
> In the revised version, we have clearly credited ``[1]`` as providing early evidence that attention is not the primary carrier both in the introduction (lines 060-063) and Appendix A. **We also highly ``encourage`` the reviewer to check our global response (for novelty and contribution) to gain a deeper understanding of how we developed our motivation and insights.**
>
>
>
> ---
>
> ### **Weakness 4: Construction of poisoned / clean variants**
>
> We will clarify the construction details. Briefly here:
>
> - **Clean datasets ($D^{\text{clean}}_i$).** For each variant (i), we draw ($D^{\text{clean}}_i$) by random sampling from an instruction dataset, using different random seeds so that the clean sets for different variants are disjoint. In early experiments, we reused the same clean data across variants and found that the resulting signatures were weaker, which motivated switching to disjoint clean sets.
>
> - **Poisoned datasets ($D^{\text{pois}}_i$).** For each ($D^{\text{clean}}_i$), we construct ($D^{\text{pois}}_i$) by inserting a variant-specific trigger and behavior. The triggers are randomly chosen tokens/phrases; the behaviors are *task-specific*, for example, if the model is deployed for sentiment steering, it can be any related content. such as "it's good or bad". We will add a more detailed description.
>
> ---

---

> ### Author Response · Authors · 2025-11-16
> **Responses - Part II**
>
> ### **Weakness 5: Coding utility metrics**
>
> We report code-generation performance for Code-LLaMA (e.g., on standard coding benchmarks) before and after purification as follows
>
> | Model                  | Phase   | pass@1 | pass@2 | pass@8 | pass@16 | pass@32 | pass@64 |
> |------------------------|---------|--------|--------|--------|---------|---------|---------|
> | CodeLLaMA-7B-Instruct  | Before  | 4.88   | 9.42   | 19.63  | 25.08   | 29.25   | 31.10   |
> | CodeLLaMA-7B-Instruct  | After   | 4.83   |   9.44    |   19.89    |   24.88     |   30.21     |   30.79     |
> | CodeLLaMA-13B-Instruct | Before  | 8.54   | 13.55  | 25.22  | 30.62   | 35.35   | 39.63   |
> | CodeLLaMA-13B-Instruct | After   |   8.47    |   13.13    |   25.46    |   30.04     |   35.79     |   39.56     |
>
> ---
>
> ### **Weakness 6: ASR not always <5\%**
>
> Our text doesn't suggest that ASR is *always* below 5\%, instead, we state that our method “frequently” reduces ASR to below 5\%. ``Line (417-418)``
>
> ---
>
> ### **Weakness 7: Paper structure**
>
> Our techniques are built directly on top of the sanity-check findings, so we placed them together in the methodology section to streamline the exposition. We will reorganize the paper so that the key findings from sanity checks appear in a separate section immediately before the methodology, and refer back to them when introducing the signature extraction and purification pipeline.
>
> ---
>
> ### **Response to Questions**
>
> **Q1: How would one suspect a model is backdoored in the first place?**
>
> Our method is designed to be applied even if the model is in fact clean, as it does not harm utility.
>
> **Q2: How are poisoned and clean variant datasets constructed, and how is diversity ensured?**
>
> Please refer to the clarification under Weakness 4.
>
> **Q3: Do backdoor signatures transfer across models/architectures?**
>
> Generally, the shared backdoor signature relies on both the architecture and the specific weights. While fine-tuned models from the same base may share some structure, it is more accurate and reliable to extract the backdoor signature directly from the given suspicious model. Our main pipeline, therefore, performs extraction per model; we will make this clear.
>
> ---
>
> ### **References**
>
> [1] Analyzing And Editing Inner Mechanisms Of Backdoored Language Models. Lamparth et al.
>
> [2] Backdoor learning: A survey. Li et al.
>
> [3] A survey of recent backdoor attacks and defenses in large language models. Zhao et al.

---

> ### Author Response · Authors · 2025-11-24
>
> Dear Reviewer Khsx,
>
> We understand that you may have a busy schedule in this period, but we would truly appreciate it if you could take a moment to review our responses (**especially for Weakness 3 as you highlighted**) and let us know if they satisfy your questions. **It would mean a lot to us as both you, as reviewers, and us, as authors, have spent a lot of effort on this**. If you have further concerns, please let us know.
>
> Thank you for your time and effort again.
>
> Authors,

---

> ### Comment · Reviewer_Khsx · 2025-11-28
>
> Dear authors,
> Thank you for replying to my concerns and questions and uploading a revised PDF of the paper.
>
> > “Safe conditions” is immediately specified by the subsequent text (lines 51–52)
>
> Based on the current write-up, I couldn't find the specification of "safe conditions". I currently do see it mentioned in lines 54-55 but I don't see an explanation of them right after. Can you quote the text here that you're referring to?
>
> > Our text doesn't suggest that ASR is always below 5%, instead, we state that our method “frequently” reduces ASR to below 5%. Line (417-418)
>
> I still don't understand this claim, but maybe I'm misinterpreting Table 2. In Table 2 (I'm referring to the bold values here), four average ASR scores for your approach are above and four scores are below 5%. Looking at the individual, non-aggregated scores in each of the respective columns of your method, there are actually more instances where ASR is >5%. So the claim that they "frequently" reduce ASR below 5% still doesn't feel truthful, unless you're referring to different data or I'm misunderstanding the presented table.
>
> > By “generative LLM” we mean instruction-following chat LLMs (e.g., LLaMA-2-Chat, Mistral-Instruct), as opposed to small text-completion models such as GPT-2. We have updated the footnote on the first page to clarify it.
>
> This seems like an arbitrary terminology change to me; GPT-2 is as much an LLM as the other models you mentioned. I don't quite understand how the ability to follow instructions is relevant to your analysis and why it would warrant a subcategorization of LLMs into instruction-following and non-instruction-following models. Could you help me understand the relevance of this subcategorization?
>
> > "[1]'s analysis is activation-based (mean ablation, logit lens, causal patching, PCP) and requires access to internal activations on data containing the trigger. Our method is purely parameter-space: we only use weight updates (such as LORA updates), which are more direct."
>
> I have read both the global response and the comment to reviewer rkEg and I agree with the latter that requiring access to weights vs. requiring activations is not a strong differentiator or advantage of your method; both require the same level of IP revelation by a company (the weights); the activations can be easily produced once you have access to the weights.
>
> I agree with the authors that the purification framework given unknown triggers is the main contribution of the paper and should be framed as such but the contribution regarding how backdoor behaviors are encoded in generative models seems overstated in the context of previous work. On that note, I would strongly suggest moving the writing in Appendix A into the actual related work section because right now, the main text doesn't give sufficient context on these prior lines of work. Putting Table 5 into the main text isn't necessary in my opinion in light of space constraints (and I'd say you can delete it from the paper and just summarize the differences in the related work section).
>
> > Weaknesses 1, 4, 5, 7
>
> I appreciate the clarifications. Thank you for taking the time to responding to and addressing my concerns.

---

> ### Author Response · Authors · 2025-11-28
> **Follow-Up Question Response**
>
> Thanks for the thoughtful comments and concrete suggestions again. **We are pleased that we have resolved the concerns: 1, 4, 5, and 7**, and respond to the rest and new concerns point-by-point below.
>
> ---
>
> **Question 1: “Safe conditions”**
>
> By “safe conditions” we specifically mean using standard, widely adopted fine-tuning hyperparameters rather than unusually aggressive ones.
>
> > (e.g., 1e-5 for full-parameter tuning and 2e-4 for LoRA adapters).
>
> Concretely, this refers to learning rates such as $1\times 10^{-5}$ for full-parameter tuning and $2\times 10^{-4}$ for LoRA adapters. We observed that in some prior work, a significant part of the reported effect actually comes from extremely large learning rates rather than from the proposed technique itself (see ``Appendix D.2``). In the revised version, we have explained this directly (``lines 053``) instead of referring to “safe conditions.” Thank you for your comment.
>
> ---
>
> **Concern 1: “Frequently below 5\% ASR”**
>
> You are right that using **frequently** with a hard **5%** threshold was not precise. When we wrote this, we only considered average scores and did not provide a detailed summary of all individual entries. After re-examining the results, we decided to rephrase the claim in terms of \emph{relative reduction} rather than an absolute 5\% cutoff. Across **all** cases, our method reduces ASR by more than **60%**; in **90%** of cases the reduction exceeds **70%**, in **87.5%** of cases it exceeds **80%**, and in **60%** of cases it exceeds **90%**. **``We therefore change the wording to say that our method “frequently reduces ASR by more than 80% relative to the attacked model,” which matches the numbers in Table 2.``**
>
> ---
>
> **Concern 2: “Generative LLMs” vs GPT-2**
>
> **To address the concerns of all reviewers, we removed the ``footnote`` and simply replace ``“generative LLMs”`` with ``“instruction-tuned LLMs”`` in the title and main text.** We do not intend to claim that GPT-2 is not an LLM.
>
> **The relevance of this distinction is reflected in the different conclusions**: Lamparth et al.~[1] find that ``early`` MLP layers plus ``trigger embeddings`` encode the backdoor in GPT-2 (not instruction-tuned), while we find that in instruction-tuned models such as LLaMA,  ``all`` MLP blocks can encode the trigger–behavior association and any subset of updated MLPs can ``re-activate`` the backdoor.
>
> ---
>
> **Concern 3: Weights vs activations / PCP wording**
>
> We apologize for the confusion caused by our wording (We guess the reason is that we mentioned “PCP” in parentheses). **Our goal was ``not`` to compare the purification techniques, but only to contrast the ``localization techniques``: their analysis uses ``activation-based tools``, while our ``sanity checks`` operate purely on ``weight updates``.**
>
> **We brought this up because there was a serious concern that the reviewer might think we had copied their localization method.** We wanted to clarify that our localization experiments were designed independently and using different tools, not to claim that our approach is superior or requires weaker access. We will correct this phrasing in the paper and remove any implication that “weights vs activations” is a practical advantage.
>
> ---
>
> **Concern 4: Moving Appendix A content**
>
> We agree with your suggestion. In the updated version, we have moved the relevant discussion from **Appendix A** into the main related work section and removed **Table 5**. The differences to prior attack and mechanistic work are now explained directly in text, rather than via a large table in the appendix.
>
> ### **References**
>
> [1] Analyzing And Editing Inner Mechanisms Of Backdoored Language Models. Lamparth et al.

---

### Official Review · Reviewer_rkEg · 2025-11-02

**Soundness:** 2
**Presentation:** 3
**Contribution:** 1
**Rating:** 2
**Confidence:** 4

**Summary:**

The submission proposes an “immunization-inspired” purification method to remove backdoors from LLMs without knowing the true trigger or having a clean reference model by creating multiple synthetic poisoned and clean fine-tuned variants of the same base model, each with different key–behavior pairs.

The authors compute parameter update differences between (LoRa and SFT) poisoned and clean variants and identifies shared, consistently aligned channels as the backdoor signature in their experiments and introduce a two-stage purification pipeline: Suppress/reinitialize high-scoring channels in MLPs or LoRA adapters and lightly fine-tune on clean data to recover fluency.

The authors claim to provide novel insights on MLPs encoding backdoor association while attention modules are not the key driver of the mechanism, that the activation is distributed across the model and different parts of the model can learn the backdoor, even when shuffled. They demonstrate the effectiveness if their method across multiple LLMs (LLaMA-2, Mistral) and attack types (e.g., BadNets, CTBA, Sleeper), outperforming their chosen baselines like pruning and fine-pruning and report results using attack success rate (ASR) and general benchmark utility. They also observes that backdoor activation is redundant and order-invariant across many MLP layers.

**Strengths:**

The submission
* proposes a novel, reference-free purification framework that extracts a shared backdoor signature across synthetic poisoned variants via magnitude + alignment scoring.
* introduces a two-stage purification pipeline (channel suppression + light clean fine-tuning) effective for both full-model and LoRA-only access.
* demonstrates good empirical results across multiple large models (LLaMA-2, Mistral) and diverse attack types, outperforming established baselines such as pruning and fine-pruning.
* provides clear experimental methodology and presentation, including ablation studies on model components and purification stages.

**Weaknesses:**

## Weakness 1 [Significance/Originality]

The paper's related work omits several critical contributions that shape today’s LLM backdoor landscape, including attacks via instruction tuning, attacks in other training steps, attacks in PEFT/LoRA settings, and, crucially, recent mechanistic analyses of how and where backdoors are encoded that come to similar findings as this submission, raising significant concerns regarding novelty and quality of the contributions made to the field.

For example, the authors did not cite influential papers in the field of backdoored LMs like

Universal Jailbreak Backdoors from Poisoned Human Feedback. Rando et al.
Trojaning Plugins of Large Language Models. Dong et al.
Attention-Enhancing Backdoor Attacks Against BERT-based Models. Lyu et al.
Poisoning Language Models During Instruction Tuning. Wan et al.
PPT: Backdoor Attacks on Pre-trained Models via Poisoned Prompt Tuning. Du et al.
Anti-Backdoor Learning (ABL): Training Clean Models on Poisoned Data. Li et al.
Blind Backdoors in Deep Learning Models. Bagdasaryan et al.
Spinning Language Models: Risks of Propaganda-As-A-Service and Countermeasures. Bagdasaryan et al.

limiting the rigor and completeness of the paper's threat model and contextual grounding.

Further, the paper

Analyzing And Editing Inner Mechanisms Of Backdoored Language Models. Lamparth et al. (Arxiv 2023, published 2024)

(which is also not cited) makes several key contributions that significantly overlap with the claimed novel insights made by this submission. In particular,
it
* identifies that early-layer MLPs and embedding projections encode backdoor behavior; attention modules are not triggers.
* introduces a method to localize, remove, or reinsert backdoor mechanisms (in clean and backdoored LLMs).
* shows backdoor activation distributed across early layers and scalable by parameters edits.
* studies the effect of keeping MLPs and attention modules fixed during fine-tuning to reduce backdoor eprformance without harming utility.
in a trigger-agnostic way (manipulation of backdoors without needing to know the trigger, only a large dataset containing it). Meaning that both show MLPs encode the malicious association while attention mainly amplifies or maintains coherence, confirm activation is distributed across layers (strongest in early MLPs), attacks are trigger-agnostic in approach (although with different methods; dataset activations vs synthetic variant contrasts), and enable backdoor removal without external clean reference models.

Besides using newer models and attack methods that since came out compared to the old paper,  this seemingly reduces the novel contributions of the submission to their method to collect the backdoor signature extraction and for the purification pipeline, studying PEFT settings, and applications like Coding.

A more rigorous literature review and better positioning of the submitted paper could strengthen it terms of significance/originality.

## Weakness 2 [Quality]

The submission sutdies attack success rate and a general utility score, but omits the metric of accidental trigger rate (ATR), which can lead to underestimating false positives or collateral damage from purification not captured in the utility score. It is also unclear how potential over-purification could be a problem, asfull-channel reinitialization may remove benign semantics or have other unmeasured side effects.

Additional experiments and clarifications could strengthen the paper in terms of quality.

## Weakness 3 [Quality]

It is unclear how real attacker backdoors may behave differently to the author self-generated poisoned variants. Also, there seems to be a dependence on the behavior knowledge of the backdoor, as “trigger-agnostic” still assumes ability to define synthetic behaviors and fine-tune models.

Additional experiments or clarifications could strengthen the paper in terms of quality.

**Questions:**

* Is there a reason ATR was not studied and can over-purification be a problem?

* How realistic are the generated attacks compared to real-world attacks?

---

> ### Author Response · Authors · 2025-11-16
> **Responses - Part I**
>
> Thank you for the careful review and for the concrete pointers to related work. Below we respond to each weakness and question in turn.
>
> ---
>
> ## **Weakness 1: Significance / Originality**
>
> **On the missing attack references.**
>
>
> - We denote the eight references suggested by the reviewer as ``[1]–[8]``.
> - ``[6]`` and ``[7]`` are conducted in CV settings; ``[3]`` and ``[5]`` focus on text-classification models; and `` [4][8]`` are already cited in our domain. In contrast, our work focuses on generative LLMs (LLaMA-2, Mistral, Code-LLaMA) with 7B–13B parameters and instruction following ability.
> - Papers ``[1], [2], [3], [4], [5], [7], [8]`` all focus on attack methodologies, while only ``[6]`` targets the defensive perspective (which is also our task of interest), but still focuses on the Computer Vision task.
> - We already cited nearly 20 attack papers, including typical attacks on **CV** and **text-classification** models, early work on **seq-to-seq** models, and recent **instruction-tuning** backdoors (``lines 095-113``), and we intentionally allocate more space to defense-related work (``lines 114-130``).
> - We have extended the related-work section (``lines 095-132``) in the revised version to discuss ``[1], [2], [3], [5], [6], [7]`` as part of the broader LLM backdoor landscape for better completeness and positioning.
>
> **Positioning w.r.t. [9] (Analyzing and Editing Inner Mechanisms of Backdoored LMs).**
>
> We agree that this is an important mechanistic study and we will explicitly cite and acknowledge it. However, our work is independent and differs in several key aspects:
> - **Model scale and setting.** ``[9]`` Study GPT-2 toy and medium models (<=355M) for text completion. We target 7B–13B *chat* LLMs (LLaMA-2, Mistral, Code-LLaMA), where behavior is instruction-response mode.
> - **Methodology.** ``[9]``'s analysis is activation-based (mean ablation, logit lens, causal patching, PCP) and requires access to internal activations on data containing the trigger. Our method is purely parameter-space: we only use weight updates (such as LORA updates), which are more direct.
> - **Mechanistic conclusions to MLP.** ``[9]`` emphasizes **early-layer MLPs + trigger embedding changes**. Our sanity checks show that in large chat LLMs, *all* MLP blocks can encode the trigger–behavior association; any subset of updated MLPs can re-activate the backdoor, even when updates are shuffled. We also discuss in a stronger setting where the trigger embeddings are kept fixed.
> - **Role of attention.** ``[9]`` concludes that attention mainly maintains language coherence in GPT-2. We find that in modern chat LLMs, attention additionally amplifies and routes trigger signals (especially in Mistral), while MLPs encode the association.
> - **Goal and threat model.** ``[9]'s`` PCP ablations are a mechanistic *editing* tool (insert/scale backdoors) and require datasets containing the true trigger. Our contribution is a **practical purification framework** that (i) does not require the attacker’s trigger to appear in any data, (ii) works under both full-model and LoRA-only access, and (iii) demonstrates large ASR reductions on modern LLMs while preserving utility.
>
> In the revised version, we have clearly credited ``[9]`` as providing early evidence that attention is not the primary carrier both in the introduction (lines 060-063) and Appendix A. **We also highly ``encourage`` the reviewer to check our global response (for novelty and contribution) to gain a deeper understanding of how we developed our motivation and insights.**
>
> ---

---

> ### Author Response · Authors · 2025-11-16
> **Responses - Part II**
>
> ## **Weakness 2: ATR and Over-purification**
>
> ATR is essential in *attack-oriented* work, where the attacker must show the backdoor remains dormant on benign inputs. For defense work, the mainstream approach, such as ``[10,11,12,13]``, didn't consider it as a metric. Indeed, we directly measure utility on **10 close-ended benchmarks** (BoolQ, ARC-C/E, GSM8K, MMLU, etc.) and **MT-Bench**. If purification were over-aggressive and produced frequent accidental activations on benign prompts, this would necessarily have degraded task accuracy or dialogue quality. Empirically, our purified models match or slightly exceed the clean models on these benchmarks while reducing ASR by large margins, indicating that collateral damage is small in practice.
>
> ---
>
> ## **Weakness 3: Realism of Attacks and “Trigger-Agnostic” Assumption**
>
> **On the realism of backdoors.**
> The synthetic variants we construct are *not the final* evaluation setting; they serve only as “vaccines” to learn the shared association signature on the suspicious model. All main results are reported on **realistic attacks implemented in BackdoorLLM** (BadNets, CTBA, MTBA, Sleeper, VPI) plus code-injection backdoors on Code-LLaMA. Moreover, our cross-attack experiment shows that signatures learned from BadNet variants transfer to CTBA/MTBA/Sleeper/VPI and substantially reduce ASR, indicating that the extracted signature captures a general trigger–behavior association mechanism rather than idiosyncrasies of a single synthetic attack.
>
> **Clarifying “trigger-agnostic”.**
> Our method is agnostic to the **attacker’s actual trigger and behavior**: we never use the true trigger tokens or poisoned data used by the attacker, and our synthetic keys/behaviors are constructed by the defender and are disjoint from the attacks used for evaluation. In real-world deployments, LLMs are typically task-specific (e.g., sentiment analysis, assistance, code generation), so the defender can reasonably hypothesize *behavior families* (sentiment shift, forced refusal, malicious snippet insertion) without knowing the attacker’s concrete phrasing or trigger pattern. Our experiments, including the cross-attack study, demonstrate that task-level synthetic behaviors are sufficient to learn signatures that generalize to unseen real-world attacks.
>
> ---
>
> ## **Questions**
>
> **Q: Is there a reason ATR was not studied, and can over-purification be a problem?**
>
> A: Please look at the response to weakness 2.
>
> **Q: How realistic are the generated attacks compared to real-world attacks?**
>
> A: Please look at the response to weakness 3.
>
> ---
>
> ## **References**
>
> [1]Universal Jailbreak Backdoors from Poisoned Human Feedback. Rando et al.
>
> [2]Trojaning Plugins of Large Language Models. Dong et al.
>
> [3]Attention-Enhancing Backdoor Attacks Against BERT-based Models. Lyu et al.
>
> [4]Poisoning Language Models During Instruction Tuning. Wan et al.
>
> [5]PPT: Backdoor Attacks on Pre-trained Models via Poisoned Prompt Tuning. Du et al. Attack.
>
> [6]Anti-Backdoor Learning (ABL): Training Clean Models on Poisoned Data. Li et al.
>
> [7]Blind Backdoors in Deep Learning Models. Bagdasaryan et al.
>
> [8]Spinning Language Models: Risks of Propaganda-As-A-Service and Countermeasures. Bagdasaryan et al.
>
> [9]Analyzing And Editing Inner Mechanisms Of Backdoored Language Models. Lamparth et al.
>
>
> [10] Crow: Eliminating backdoors from large language models via internal consistency regularization. Nay Myat Min et al. [ICML 2025]
>
> [11] Defense against Backdoor Attack on Pre-trained Language Models via Head Pruning and Attention Normalization Xingyi Zhao et al. [ICML 2024]
>
> [12] NEURAL ATTENTION DISTILLATION: ERASING BACKDOOR TRIGGERS FROM DEEP NEURAL NETWORKS [ICLR 2021]
>
> [13] Pre-activation Distributions Expose Backdoor Neurons. Runkai Zheng et al. [NeurIPs 2022]

---

> ### Author Response · Authors · 2025-11-24
>
> Dear Reviewer rkEg,
>
> We understand that you may have a busy schedule in this period, but we would truly appreciate it if you could take a moment to review our responses and let us know if they satisfy your questions. **It would mean a lot to us as both you, as reviewers, and us, as authors, have spent a lot of effort on this**. If you have further concerns, please let us know.
>
> Thank you for your time and effort again.
>
> Authors,

---

> ### Comment · Reviewer_rkEg · 2025-11-27
>
> Dear authors,
>
> Thank you for your comments (general and specific to my review) and uploading a rebuttal revision with highlighted changes!
> Some concerns have been partially addressed, while most concerns still need to be resolved. See below for details.
>
> ### Weakness 1
>
> * Why should a backdoor behavior be different in an instruction tuned or larger model, beyond being more distributed across layers? Generally speaking, scaling up a known phenomenon is not a novel insight, unless new qualitative behaviors appear, but an incremental result. The claimed “new” mechanistic facts (redundancy, shuffling tolerance) sound more like extensions, not contradicting or displacing findings on smaller, non-instruction-tuned models. The proposed redefinition of “generative LLM” in the rebuttal revision to specifically only refer to _“large instruction-tuned models (e.g., LLaMA-2-Chat family, Mistral-Instruct family), rather than small GPT-2–style completion models or seq2seq encoders.”_ and labeling GPT2 intentionally not as generative LM but as _"small text-completion models"_ feels misleading.
>
> * Why does a parameter-space methodology require less access than an activation-based methodology? For the poisoned and clean model variants in your methodology, you also need full model access (in security terms, training access is a stricter threat model than inference access). I see that there is a difference for the LoRA results, but not for the fine-tuning results. Also, both methodologies need a dataset to make sense of either the parameters or the activations. Essentially, different methods finding the same insight does not make the insight new, which is why I am arguing to reframe them as a confirmations, but remove them as contributions.
>
> * Could you clarify the distinction of the role of attention in your rebuttal comments (to me and general)? Comparing the two works, "attention mechanism is still required to route the trigger signals, while encoding the association itself?" feels accurate for both.
>
> * This is an excellent point you should highlight as contribution, as it does not confirm prior insights. “[9] emphasizes early-layer MLPs + trigger embedding changes. Our sanity checks show that in large chat LLMs, all MLP blocks can encode the trigger–behavior association; any subset of updated MLPs can re-activate the backdoor, even when updates are shuffled.”
>
> Thus, my originality and significance concerns have not been addressed and the offered argumentation and rephrasings feel misleading or even potentially disingenuous.
>
> ### Weakness 2
>
> I appreciate your comment regarding ATR being attack-oriented metric and that general utility metrics indirectly reflect ATR effects, but an explicit measurement would strengthen the claim that over-purification does not harm benign behaviors. Benchmark accuracy is a coarse proxy by design and it cannot detect subtle or localized over-purification (e.g., benign phrases semantically similar to triggers). Thus, I do not think my concern has been addressed.
>
> You can essentially verify this by checking whether the purified model accidentally activates the backdoor behavior on benign prompts that do not contain the trigger (or contain only “trigger-like” patterns). Can you clarify why solely using benchmarks scores guarantee such checks are not needed?
>
> ### Weakness 3
>
> Thank you for your comments; they address my second concern on the meaning of “trigger-agnostic” sufficiently well! However, I still have concerns regarding the first part on transferring the author generated poisoned variants to real attacker backdoors.
>
> Can you provide evidence that synthetic updates produce parameter-space shifts that resemble the ones induced by real attacker fine-tuning? The showed cross-attack transfers demonstrate that the extracted signatures helps remove backdoors, but not that the learned signature is causally the same as the real attacker-induced mechanism or that the synthetic variant updates faithfully approximate real backdoor updated geometry. Can you comment on that?
>
> Thank you!

---

> ### Author Response · Authors · 2025-11-28
> **Follow-Up Question Response Part I**
>
> Thanks for the thoughtful comments and concrete suggestions. **We are pleased that we have resolved part of your concerns,** and we respond to the rest and new concerns point-by-point below.
>
> ---
>
> ### **Weakness 1**: Originality and Significance
>
> **Concern 1**
>
> - We initially viewed our MLP-related finding as novel in our first-round rebuttal because, for 7B–13B models, if we directly follow the insight of **Lamparth et al. [1]** and remove the poisoned updates in early layers and the embedding changes, we would have expected the backdoor behavior to disappear. **``However, in our setting, we find that even after doing this, the backdoor behavior can still be stably activated. In this sense, our conclusion is not fully consistent with the previous work``**, and we hope the reviewer can understand our standpoint in the first-round rebuttal. That said, we now understand the reviewer’s underlying concern, and in the second revised PDF we will **explicitly state that our analysis is extending** Lamparth et al. [1] to a new regime and **adding additional observations on top of it**.
>
> - The reviewer may have misunderstood our intention with the footnote on “generative LLMs”. It was not added in response to this review, nor was it meant to emphasize a difference from **Lamparth et al. [1]**. It was added to address another reviewer’s comment that our definition of “generative LLM” was unclear. **``Also, we did not “redefine” the term; we only clarified that in this paper, “generative LLMs” specifically refer to relatively large instruction-finetuned chat models such as LLaMA-2-7B-Chat``**. **To avoid any further confusion**, we remove the footnote and change the title and the narrative from ``generative LLMs`` to ``instruction-tuned LLMs``. We hope the reviewer will not misinterpret the original footnote as an attempt to dodge prior work or to be disingenuous.
>
> **Concern 2**
>
> Regarding the question of why our method would “require less access” than an activation-based methodology, **there is a misunderstanding of the context**. In that part of the 1st round rebuttal, we were not discussing the entire purification framework, **``but specifically contrasting the localization procedure of Lamparth et al. [1] with our own sanity checks``** (We apologize that we use **``PCP``** at that point, it is an unintentional mistake. We guess that's the reason the reviewer misunderstood). In this narrow context, their analysis validates mechanisms in activation space, whereas our sanity checks operate purely in parameter space by focusing on weight updates.
>
> Our goal there was only to highlight that the techniques used for **localization are different**, in order to clarify that our sanity-check study was **conducted independently rather than copying the original method**. For the broader discussion of whether the resulting insight is “the same or new”, please refer to our previous response on Weakness 1.
>
>
> **Concern 3**
>
> Regarding our characterization of attention, what we originally wanted to emphasize is the word **``“amplify”``**. In the submitted main text, we used **``“amplify”``** to describe the role of attention, and we agree that **"route the trigger signals”** is essentially the same in both our work and Lamparth et al. [1]. We removed **"route"** from the comparison table in the global response.
>
> The use of **“route”** in the rebuttal was written under time pressure during the rebuttal, and we did not carefully distinguish it from our original wording— this was an unintentional error on our side. We apologize for the confusion, and hope the reviewer can understand that our intention was **only to highlight the amplification** role of attention.
>
> **Confirmation 4**
>
> We thank the reviewer for their confirmation. In the revised PDF, we have already updated the text to explicitly acknowledge the contribution of Lamparth et al. [1]: they were the first to show, in the backdoor context, that MLPs are more important than attention. **On top of this, our work focuses on the additional phenomenon you suggested**: “*``[9] emphasizes early-layer MLPs + trigger embedding changes. Our sanity checks show that in instruction-tuned LLMs, all MLP blocks can encode the trigger–behavior association; any subset of updated MLPs can re-activate the backdoor, even when updates are shuffled``.*” We agree that this is the right way to frame our mechanistic contribution and highlight this distinction more clearly in the revised version.
>
> ### **References**
>
> [1] Analyzing And Editing Inner Mechanisms Of Backdoored Language Models. Lamparth et al.

---

> ### Author Response · Authors · 2025-11-28
> **Follow-Up Question Response Part II**
>
> ### **weakness 2**: On ATR and over-purification
>
> We thank the reviewer for further clarifying their concerns. As the reviewer notes,  **accidental trigger rate** (ATR) is typically used in **attack** work to verify that a backdoor does not accidentally fire on benign inputs. In our experiments, when implementing the backdoor attacks baseline, we already check this property: for each backdoored model, we evaluate **ATR** on a held-out set of **500 benign prompts** randomly sampled from the Alpaca dataset, ensuring that none of them contain the trigger tokens. We report the results for **LLaMA-2-7B-Chat** and **LLaMA-2-13B-Chat** under the **sentiment steering** task with five attack methods (**BadNets, VPI, Sleeper, MTBA, CTBA**) as follows: In all cases, the **ATR** of the attacked models is **0%** within our sampling resolution. Our implementations for the attack baseline and evaluation code follow the publicly released repository, and these numbers are fully reproducible.
>
> Following the reviewer’s suggestion, we also recompute ATR on the **purified** models using a freshly resampled set of 500 benign prompts (**again from Alpaca, without triggers**). We find that the ATR of the purified models remains **0%** in all these settings as well. Table 1 summarizes these results.
>
>
> **Table 1. Accidental trigger rate (ATR, %, lower is better) on 500 benign Alpaca prompts**
>
> | **Model**          | **Attack** | **Attacked** | **Purified (Ours)** |
> |--------------------|-----------|--------------|----------------------|
> | **LLaMA-2-7B-Chat** | BadNets   | 0.0          | 0.0                 |
> |                    | VPI       | 0.0          | 0.0                 |
> |                    | Sleeper   | 0.0          | 0.0                 |
> |                    | MTBA      | 0.0          | 0.0                 |
> |                    | CTBA      | 0.0          | 0.0                 |
> | **LLaMA-2-13B-Chat**| BadNets   | 0.0          | 0.0                 |
> |                    | VPI       | 0.0          | 0.0                 |
> |                    | Sleeper   | 0.0          | 0.0                 |
> |                    | MTBA      | 0.0          | 0.0                 |
> |                    | CTBA      | 0.0          | 0.0                 |
>
>
> We also clarify why we previously argued that benchmark performance implicitly constrains **ATR**. Take **GSM8K** as an example: it consists of benign math word problems with essentially a correct numerical answer per instance. For example:
>
> > **Question:** James decides to run 3 sprints 3 times a week. He runs 60 meters each sprint. How many total meters does he run a week?
> > **Answer:** 540
>
> Here, only the model answering with **540** is counted as correct, and any backdoor-style behavior (e.g., refusals, toxic text, or unrelated content) would be marked wrong. If a purified model had a high **ATR**, it would frequently switch from producing such numerical solutions to generating backdoor-like outputs even on these benign prompts, which would substantially degrade **GSM8K** accuracy. Instead, we observe that our purified models closely match (or slightly exceed) the clean models on **GSM8K** and other benchmarks, and the explicit **ATR** results above are consistent with this: in our setting, both attacked and purified models exhibit effectively zero accidental triggering on benign prompts.
>
> ---
>
> ### **Weakness 3:** New Question and Evidence Request.
>
> We understand this concern, and we agree that a more direct parameter-space comparison between synthetic and real backdoor updates would be informative. Having said that, this experiment requires additional careful setup, and we are still in the process of designing and running it. Given the tight rebuttal timeline, we prioritized addressing the other main concerns first in the revised PDF. We will post the results by the rebuttal deadline.
>
> **[Update For AC]**: Please refer to our confidential message regarding how we design the experiment to provide additional evidence to satisfy the reviewer's request in weakness 3.

---

### Author Response · Authors · 2025-11-16
**To all Reviewers and AC: Global clarification on contribution and novelty**

Dear All:

Thank you for your time and valuable insights on this work. Several reviewers (rkEg and Khsx) expressed concern that our second insight (``lines 60–62``: “MLPs encode the malicious association …”) is not entirely new compared to ``[1]``, and therefore the overall contribution should be considered low to 1. We would like to clarify our position.

> **First**, our work was conducted **independently**, and our methods, model scale, and the details of the conclusions are fundamentally **different**.

| Aspect                      | Lamparth et al. [1]                                   | This paper                                                |
|----------------------------|--------------------------------------------------------|-----------------------------------------------------------|
| **Model scale & setting**  | GPT-2 toy / medium (≤355M), text completion           | 7B–13B *chat* LLMs (LLaMA-2, Mistral, Code-LLaMA), instruction–response |
| **Locating Methodology (``Not purification method``)**            | Activation-based： needs internal activations on trigger-containing data | Parameter-space only: no activations required |
| **MLP-related finding**    | Backdoor mainly in **early MLP layers** plus **trigger embedding** changes | Backdoor associations redundantly encoded across **almost all MLP blocks**; remain active under block shuffling; also studied with **fixed trigger embeddings** |
| **Role of attention**      | Primarily maintains language coherence in GPT-2       | Amplifies ~and routes~ trigger signals (esp. in Mistral), while MLPs encode the association itself |
| **Goal & threat model**    | Mechanistic **editing** (insert/scale backdoors); assumes data **containing the true trigger** | Practical **purification**: no attacker trigger in any data; works under **full-model** and **LoRA-only** access; large ASR reductions with preserved utility |

> **Second**, similar observations that MLPs play a central role in storing associations have already appeared **earlier** in knowledge editing work ``[2]``. If we have to speak of the direct “influence”, then it would be the ``ICLR 2025 best paper [3]`` on hallucination and factual correction **in MLP blocks**: they operate in a setting where the target knowledge and inputs are known and explicitly edited; **in contrast**, we study backdoors under completely unknown triggers, where the defender does not know key–behavior pairs at all. We will clearly position our results as extending this intuition from knowledge-editing and hallucination correction to the backdoor setting in large generative LLMs.

> **Most importantly**, this MLP insight is only the **starting point that motivates our technique**, not the main contribution by itself. The core of our paper is a backdoor elimination framework, not a localization study: an immunization-style, reference-free signature extraction method plus a two-stage purification pipeline, evaluated on modern 7B–13B chat LLMs under both full-model and LoRA-only threat models and across multiple realistic attack families. Even if one considers the high-level insight that “MLPs matter more than attention” as shared with prior work, ``this does not diminish our contribution to “1”``: the main novelty lies in transforming this understanding into a practical, scalable defense for generative LLMs.

> **Finally**, we have acknowledged (``Introduction (lines 060-063) and Appendix A``) in the revised version that ``[1]`` has earlier pointed out, in smaller GPT-2–style models, that early MLPs are more important than attention for backdoor mechanisms. Again, please note our paper builds on an independently obtained, consistent observation and moves the focus from localization to actual purification in large-scale chat LLMs.

**References**

[1] Analyzing And Editing Inner Mechanisms Of Backdoored Language Models. Lamparth et al.

[2] Locating and Editing Factual Associations in GPT. Kevin Meng et al. [NeurIPS 2022]

[3] AlphaEdit: Null-Space Constrained Knowledge Editing for Language Models, Junfeng Fang et al. [ICLR 2025 Outstanding Paper]

Authors

---

### Author Response · Authors · 2025-11-16
**Global update on the revised PDF: all changes are highlighted in blue.**

We have uploaded a revised version of the paper, with minor changes in **blue**. The changes are deliberately minimal and do not affect the method or results:

- (1) We updated lines (060-063) and added a short subsection in Appendix A discussing prior mechanistic and knowledge-editing papers (e.g., Lamparth et al. on GPT-2).
- (2) Added more citations in the introduction and related work (**please check more details to address weakness 1 posted by reviewer, rkEg, and several from the other reviewers**).
- (3) Added a footnote on the first page to clarify generative LLM.
- (4) Deleted a confusing part in section 3.1 (lines 155-156 in the old version).

We addressed **all other points** raised by the reviewers in detail in this rebuttal and will incorporate them in the camera-ready version if this paper is accepted.

---

### Author Response · Authors · 2025-11-30
**Thank You and Next Steps**

We would like to express our sincere gratitude to everyone involved in the review process.

**To the original reviewers and AC**:

First, we deeply appreciate all reviewers for their engagement throughout the rebuttal stage.
- We are grateful for the strong recognition from **dxCf** and **1Jtm**.
- We sincerely thank **rkEg** and **Khsx** for their constructive and consistent participation during the discussion.

We also thank our previous AC for coordinating the review process before the system-wide changes.

**To the new AC:**

Second, we would like to thank our new AC for stepping in under tight time constraints. We recognize that this is a particularly challenging situation, and we truly appreciate the time you have taken to address this matter. We are currently preparing

- a revised PDF
- a concise summary

to assist the AC in understanding how we addressed the reviewers’ concerns. We will submit these materials very shortly.

**Finally**, thank you again to everyone for their time, effort, and dedication to ensuring a fair and rigorous review process.

---

### Author Response · Authors · 2025-12-01
**To AC: Global Summary**

To save the AC’s time, we briefly summarize how we addressed the concerns from reviewers with **negative** score, **rkEg** and **Khsx**.

---

**1. Main concern: relation to [1]**

1st-round rebuttal successfully let reviewers understand this work was conducted independently, and we didn't deliberately hide the connection to **[1]**.  For the remaining concerns:

* The **“no activation access”** point was **misunderstood**: We were referring only to the **``mechanism localization``** methods ( [1] use **activation-based causal** techniques, whereas we use **weight-update ablations** ), not to the final ``purification framework``.

* **No difference** because the two works describe **attention** as ``routing`` trigger signals: Our intended emphasis is only **``amplify``** instead of ``routes``. We remove the word "routes" from the comparison table in rebuttal.

* Both reviewers **ultimately agreed** that our mechanistic contribution is acceptable and **mainly asked us to polish the writing**: In response, we have
    - Moved and polished the content of **Appendix A** into the main Related Work section  (``lines 056-063``)
    - Replaced the comparison table with a text description
    - Rephrased our 1st and 2nd insights in the introduction. (``lines 138-155``)

---

**2. Concern about the term “generative LLM”**

* We deleted the footnote added in the last revision.
* We keep “generative LLMs” only in the title (to align with the original submission)
* We replace all occurrences in the main text with “instruction-tuned LLMs”.

---

**3. Concern about ATR (rkEg)**

* We reported ATR (mostly reported by attack method) = 0\% before/after purification in several settings.
* We added a GSM8K example showing that high ATR would immediately hurt benign-task accuracy.

---

**4. Concern about “frequently reduces ASR below 5\%” (Khsx)**

* We changed to “frequently reduces ASR by more than 80\% relative to the attacked model,” (``line 420``) according to the table.

### **References**

[1] Analyzing And Editing Inner Mechanisms Of Backdoored Language Models. Lamparth et al.

---

### Meta-Review · Area_Chair_sicL · 2025-12-08

**Summary:**

The paper proposes a backdoor purifying framework for instruction-tuned LLMs. The core idea is to generate synthetic backdoored variants of a suspicious model to learn a backdoor signature (consistent parameter updates in MLPs), which is then used to purify the model without access to a clean reference or the specific trigger.

Reviewers dxCf and 1Jtm highlight the practical value of the reference-free setting, the effectiveness of the method on large models, and the robust experimental results across various attack types.

Reviewers rkEg and Khsx argue that the paper’s motivating insight, that backdoors are encoded in MLPs while attention amplifies the trigger, overlaps significantly with prior work. They are concerned that the paper claimed this as a novel discovery rather than a validation in a new regime. Regarding this concern, the authors provide sufficient evidence to clarify their contribution compared with the related work and modified the paper.

In summary, I recommend this paper for acceptance.

**Reviewer Concerns:**

Most concerns raised by reviewers are resolved, especially the novelty of mechanistic insight (see summary). I list some as follows.

- Accidental Trigger Rate (ATR): Reviewers rkEg and Khsx worried about over-purification harming benign performance. The authors provided new data showing 0% ATR on benign prompts and strong performance on standard benchmarks (GSM8K, etc.), mitigating this concern.
- Terminology: The ambiguity of "Generative LLM" vs. "GPT-2" raised by Khsx and rkEg was addressed by the authors agreeing to use the more precise term "Instruction-tuned LLMs".
- Missing Metrics: The lack of coding utility benchmarks (raised by Khsx) was resolved with new tables showing preserved performance on Code-LLaMA.

**Reviewer Scores:**

Reviewer dxCf and 1Jtm confirm the author's rebuttal and likely keep the positive scores as 6 and 8.

Reviewer rkEg may raise the score from 2 to 4. This reviewer remained that the originality and significance concerns have not been addressed. They likely would not have flipped to an accept, but might acknowledge the empirical value with a slightly higher yet still negative score.

Reviewer Khsx may raise the score from 2 to 4 or 6. The reviewer engaged deeply and acknowledged that Weaknesses 1, 4, 5, and 7 were resolved. Besides, the author also responded to the reviewer's further concerns and provided clarification.

---

### Decision · Program_Chairs · 2026-01-26

Accept (Poster)